# A Pilot, Phase II, Observational, Case-Control, 1-Month Study on Asthenopia in Video Terminal Operators without Dry Eye: Contrast Sensitivity and Quality of Life before and after the Oral Consumption of a Fixed Combination of Zinc, L-Carnitine, Extract of Elderberry, Currant and Extract of Eleutherococcus.

**DOI:** 10.3390/nu13124449

**Published:** 2021-12-13

**Authors:** Gemma Caterina Maria Rossi, Luigia Scudeller, Federica Bettio, Giovanni Milano

**Affiliations:** 1Surgical Sciences Department, University Eye Clinic, Fondazione IRCCS Policlinico San Matteo, P.le Golgi 19, 27100 Pavia, Italy; f.bettiofederica@gmail.com (F.B.); gmilano@unipv.it (G.M.); 2Clinical Epidemiology and Biometric Unit, Scientific Direction, Fondazione IRCCS Policlinico San Matteo, P. le Golgi 19, 27100 Pavia, Italy; l.scudeller@smatteo.pv.it

**Keywords:** diet, lifestyle, nutraceutics, eye, VDT, video display terminal, dry eye, contrast sensitivity, quality of life, ocular surface, COVID-19

## Abstract

The aims of the study were to investigate the ability and effectiveness of an oral intake of a fixed combination of zinc, L-carnitine, elderberry extract, black currant and Eleutherococcus extract in controlling the symptoms of eyestrain in videoterminal (VDT) users and to record its effects on contrast sensitivity. A single-center, phase II, observational, case-control, 1-month study in VDT workers without dry eye disease was carried out. Demographics and number of actual hours at VDT/day were taken into account. All subjects underwent a complete ophthalmic examination, including assessment of contrast sensitivity, and completed the computer vision symptom scale questionnaire at baseline and one month later. A total of 30 Caucasian subjects adhered to the required inclusion criteria and completed the study; 15 subjects were treated (T) and 15 were controls (C). All clinical data at baseline were similar in both groups (*p >* 0.05): after one month, all subjects had stable visual acuity, refractive defect and intraocular pressure (IOP); screen exposure time was unchanged. Regarding symptoms, at randomization, the groups had a similar score: 33.1 ± 3.3 in T and 32.8 ± 5.6 in C. One month later, the computer vision symptom scale (CVSS) questionnaire score decreased by −14.1 ± 3, 1 (*p =* 0.000) and −2.3 ± 1.8 (*p =* 0.568), respectively. Regarding contrast sensitivity, in group C the values of spatial frequencies remained unchanged, while they improved in almost all the cycles per degree stimuli in the treated group. Oral intake of a fixed combination of zinc, L-carnitine, elderberry extract, black currant and eleutherococcus extract can significantly improve contrast sensitivity and symptoms in VDT workers with no signs of dry eye disease.

## 1. Introduction

Employees who spend at least 20 h per week at a video display terminal (VDT) as their main job are referred to as “video terminal workers”. Prolonged use of video terminals not only generates musculoskeletal disorders, but also causes ocular complications.

During the current COVID-19 coronavirus pandemic, much of the population spent much more time at home, being involved in digital activities across multiple devices.

“Screen time”, which can be defined as the time spent in front of the screen of computers, PDAs and mobile phones, has increased dramatically during the lockdown, so much so, that most people have used and still use at least one device for more than 6 h per day [1].

As a result, in the past two years, many people, of all ages, have developed symptoms and signs of “computer vision syndrome” (CVS), a condition introduced in 2002 to define eye disorders resulting from the use of VDT due to prolonged and sustained visual attention to a monitor and a reduced blink rate [2]. Computer work puts a strain on the eye muscles: the bright contrast between the text on the screen, on a document and the symbols on the keyboard is excessive, up to 25,000 movements per day of adaptation to light. The eye muscles are in the rest position if objects are observed at a distance > 6 m; using the PC screen, the eyes stare at very close objects for a long time. If monitors and documents to be read are not placed at approximately the same distance, the eye muscles are forced to continuously vary the focus [2].

The consequences are an overload of accommodation and convergence linked to the commitment and prolonged static position and an overload of pupillary motility and retinal adaptation due to the lighting conditions of the workstation.

Environmental conditions, such as air conditioning and low humidity, act as cofactors in the pathogenesis of “computer vision syndrome” and add their effects to the physiological ones listed above [2,3].

The problem is also: visual fatigue causes eye problems and, therefore, creates discomfort for the subject, but it is also the main cause of high levels of fatigue and errors in the work and lost working days.

A review in 2005 found that “asthenopia, glare and accommodative difficulty are all aspects of CVS”, often associated with decreased blinking and dry eye [2,4]. The dry eye observed in VDT users is mainly due to increased tear evaporation [5] and malfunction of the meibomian glands [6] which induce a vicious cycle of interactions between intrinsic and extrinsic conditions that can amplify the severity of the dry eye [7] and which can have a negative impact on the quality of vision. Vision quality is affected by the good health of the “ocular surface” system which affects contrast sensitivity.

International studies have found a high prevalence of dry eye among VDT workers, ranging from 4% to 6% (confirmed dry eye) to 40–55% (suspected dry eye) [8,9,10,11], while some VDT workers do not suffer from dry eye even if they do have eye problems related to computer use.

The main objective of the present study was to investigate the ability and effectiveness of an oral intake of a fixed combination of zinc, L-carnitine, elderberry extract, currant and Eleutherococcus extract to control the symptoms of eyestrain (through a specific questionnaire) and to improve the quality of vision (through a contrast sensitivity test) in VDT users without a diagnosis of dry eye but with symptoms of ocular discomfort.

## 2. Materials and Methods

A single-center, phase II, observational, case-control, 1-month, no-profit study was carried out at the University Ophthalmology Clinic of Pavia according to the Declaration of Helsinki after the approval of the Local Ethics Committee (prot. 2015000565). All subjects provided written, informed consent prior to enrollment.

Materials and methods are described in a previous article evaluating the prevalence of dry eye in VDT workers [11], of which the present study is a substudy.

The inclusion criteria for this study were: consecutive subjects working at the foundation as VDT workers referred to the eye clinic in order to complete an ocular status evaluation according to D.LGS 81/08 (i.e., the compulsory heath assessment of workers in Italy); without (random or other) selection; absence of dry eye; willingness to participate in the study; any gender and any age; visual acuity better than 0.9 decimal with refractive correction <± 1 D; and transparent dioptric means.

Diagnosis of dry eye was based on an ophthalmic examination as follows: presence of dry eye disease (DED) as a concomitant presence of tear film break-up time (T-BUT) < 10 s and corneal staining of any grade (i.e. > grade 0); suspected DED due to the presence of T-BUT < 10 s or corneal staining of any degree; while the absence of DED was the concomitant presence of T-BUT > 10 s and the absence of corneal staining (i.e., grade 0) [11,12].

Exclusion criteria: subjects who have undergone eye surgery in the previous 6 months or who have had systemic or ocular diseases (such as rheumatoid arthritis, Sjogren’s syndrome, rosacea, infectious diseases, previous diagnosis of glaucoma, diabetic retinopathy, optic neuritis, multiple sclerosis, pituitary adenoma) which could interfere with the state of the ocular surface or with the results of contrast sensitivity.

This sample of consecutive VDT workers who did not have dry eye disease were asked to take one tablet daily of a fixed combination of zinc, L-carnitine, elderberry extract, currant and Eleutherococcus extract (Meramirt CM^®^) in the morning in a 250 mL glass of water for one month (treated group). A similar sample, i.e., consecutive VDT workers without dry eye but not receiving the tablet, was evaluated as a “control group”.

All subjects who joined the study underwent contrast sensitivity assessment and completed a questionnaire on dry eye symptoms in VDT workers both at the time of their enrollment and after one month of follow-up.

All subjects underwent a complete ophthalmic examination that included near and far best corrected visual acuity (VA), T-BUT and corneal staining at both visits.

The following variables were collected for each subject: date of birth and gender, schooling level, socio-professional category, systemic diseases and therapies, eye diseases and therapies, use of contact lenses, years of work at a VDT, number of actual hours at VDT per day, number and hours of daily breaks.

The CVSS17 is a questionnaire containing 17 items with different rating scales. The 17 items of the questionnaire were designed to obtain information on 15 different symptoms, regarding the severity of symptoms, their frequency and the subjects’ opinion [13]. It is valid for both genders and for both presbyopic and non-presbyopic people.

All subjects were asked to self-complete the questionnaire before the visit with the ophthalmologist to avoid interference with the results of the visit (communication of bad/good news).

The same observer (G.C.M.R.) performed fluorescein staining and break-up time to minimize the variability of these evaluations.

The contrast sensitivity test measures the ability to distinguish ever finer increases in light versus dark (contrast). Contrast sensitivity plays an important role in low light, fog or glare situations or when the contrast between objects and their background is low.

The test is done with glasses or contact lenses if subjects use optical correction. The Vistech contrast sensitivity test was used in the present study. These charts consist of sine wave gratings. Each chart contains five rows and nine columns of circular photographic plates (disc) on a gray background. Each row has different spatial frequency (1.5 to 18 cpd at three meters) and the contrast within the row reduces from left to right. The gratings are presented in three orientations: vertical 90 degrees, 15 degrees clockwise or anticlockwise. Only one operator (FB) performed the assessment of contrast sensitivity; she only performed the test, not knowing whether the subjects examined were part of the control group or the treated group.

Meramirt CM^®^ is based on zinc (10 mg), L-carnitine (50 mg), fruits extract of elderberry (300 mg), currant (100 mg) and extract of Eleutherococcus root (50 mg). These components can be useful in counteracting the oxidizing action of free radicals, playing a useful role in the visual cycle. In detail, Eleutherococcus is defined as an “adaptogen”, which increases the body’s resistance to metabolic stress from overwork by internal and external agents [14,15]. Zinc is present in all types of retinal neurons, as well as in the cells of the pigment epithelium, and participates in numerous enzymatic activities of the retina, in particular of the retinal pigment epithelium, and is important in the prevention of degenerative processes of ocular tissues [16].

Carnitine favors the production of energy for the cell, transmitting the fatty acids for their metabolic use, and is useful for the metabolism of ocular muscle tissue. It has also demonstrated antioxidant properties in animal models [17]. Elderberry is rich in anthocyanins and catechins and stimulates the visual pigment, thus improving the quality of vision, especially in low light conditions [18]. The extract of the currant fruit is rich in polyphenols; it helps to maintain or restore the functionality of the capillary wall within normal levels, thus promoting the integrity of the small retinal vessels and the trophism of the retinal tissue [19].

### Statistical Analysis

The study was proposed to all subjects who met all of the inclusion criteria and none of the exclusion criteria. Descriptive statistics were produced for the demographic, clinical and laboratory characteristics of the cases. Mean and standard deviation (SD) are presented for normally distributed variables, median and interquartile range (IQR) for variables which are not normally distributed and number and percentages for categorical variables. Groups were compared with parametric or non-parametric tests, according to data distribution, for continuous variables and with a Pearson’s Chi2 test (Fisher’s exact test, where appropriate) for categorical variables.

In all cases, two-tailed tests were used. The cut-off of significance for the *p* value was 0.05.

## 3. Results

A total of 30 Caucasian subjects meeting all inclusion criteria and none of the exclusion criteria agreed to participate in the study and signed informed consent. Fifteen were randomized to group T = treatment (daily intake of one tablet of the fixed nutraceutical combination) and 15 to group C = control.

The demographics are summarized in
Table 1.

Briefly, most subjects (23.77%) were women; all subjects were employed at IRCCS Policlinico San Matteo Foundation; and schooling level was good with high school graduation achieved by 14 (47%) and a bachelor’s degree by 6 (20%).

About half of the subjects were presbyopes (16.53%), while about one in four used glasses for distance correction (8.26%). Visual acuity was high (0.97 ± 1.1 decimals). Mean IOP was 14.5 ± 1.3 mmHg. All clinical data were similar in both groups at baseline examination (*p >* 0.05).

During the 1-month study, all participants had stability of: visual acuity, refractive defect, IOP and screen exposure time.

Figure 1 reports data on changes in symptoms examined with the CVSS questionnaire over time. At enrollment, both groups had a similar score: 33.1 ± 3.3 in the treated and 32.8 ± 5.6 in the control. One month later, the computer vision symptom score was reduced by −14.1 ± 3, 1 (*p =* 0.000) and −2.3 ± 1.8 (*p =* 0.568), respectively.

With regard to contrast sensitivity, in the control group the values of the spatial frequencies remained substantially unchanged, while they improved in almost all the cycles per degree stimuli in the group treated with the tablet (Table 2).

## 4. Discussion

The present study showed that the oral intake of a fixed combination of nutraceuticals with specific effects on vision (zinc (10 mg), L-carnitine (50 mg), elderberry extract (300 mg), black currant (100 mg) and Eleutherococcus root extract (50 mg)) can improve the quality of vision examined with contrast sensitivity and also has a positive impact on the symptoms of computer vision syndrome.

We decided to use a validated and reliable questionnaire, the CVSS, to investigate the symptoms related to computer vision syndrome and to quantify their impact on quality of life. Previous literature has used this questionnaire and has noted that the use of a VDT for many hours negatively affects the quality of life associated with vision; in particular, the level of discomfort appears to increase with the amount of digital screen use and with the onset of dry eye [11].

For two main reasons, in the present study, it was decided to examine only those subjects without ocular surface alteration, i.e., those without dry eye disease, but who reported symptoms of visual discomfort due to the use of VDT for work: (1) the symptoms themselves affect fatigue and errors in work, affect the quality of work and can even lead to lost working days; (2) tear film debris, conjunctival staining, corneal staining and Schirmer’s test score are not significantly associated with contrast sensitivity [20].

Visual acuity is usually assessed during a routine ocular examination, but contrast sensitivity plays a crucial role in quality of vision. Ocular surface disorders and conditions that lead to dry eye, such as prolonged use of a VDT, can affect visual function by altering both visual acuity (epithelial keratitis or ocular inflammation due to dry eye) and quality of vision (instability of the ocular surface). A recent article [20] found that severe meibomian gland obstruction was significantly associated with worse mean log contrast sensitivity and that the degree of abnormality in meibomian gland secretions was significantly associated with worse mean log contrast sensitivity, while longer tear film break-up time was significantly associated with better mean log contrast sensitivity.

Contrast sensitivity is the ability of the eye to detect small variations in illumination on targets that have no well-defined limits defining the threshold between visible and invisible [21]; therefore, it reflects the quality of vision and represents a measure as important as visual acuity.

Taking a fixed combination of zinc, L-carnitine, elderberry extract, black currant and Eleutherococcus extract for one month improved contrast sensitivity at higher spatial frequencies. Age is known to affect contrast sensitivity, resulting in a linear decline for mid and high spatial frequencies, possibly due to changes in lens spherical aberration, but sensitivity for low spatial frequencies is independent of age [22].

Contrast sensitivity is also altered when the refractive defects are not corrected: for mild refractive defects there is a decrease first to smaller grating sizes or to higher spatial frequencies; for moderate and severe refractive defects, the contrast sensitivity decreases in the middle and then in the larger grating sizes (middle to lower spatial frequencies).

Our sample consisted of subjects with very good visual acuity and low refractive disorder (no more than +/− 1D, as per the eligibility criteria) and no subjects wore contact lenses; therefore, the improvement in contrast sensitivity at higher spatial frequencies observed in our subjects was independent of the correction of refractive errors and could be explained by the intake of nutraceuticals. It seems that the intake of these nutrients improved the refractive condition of the studied sample.

At present, it is well known from in vivo and in vitro studies that anthocyanins and flavonoids are widely distributed in ocular tissues after crossing both the blood–aqueous barrier and the blood–retinal barrier [23,24]. This evidence could explain our data as a positive effect of the antioxidant effects of these nutrients, leading to improved and increased blood circulation in the retinal capillaries and to production of retinal pigments that improve vision.

Our observations are also in agreement with the study by Nakaishi; in a double-blind, placebo-controlled crossover study examining the human intake of black currants, authors found that daily intake of anthocyanins improved dark adaptation, symptoms of asthenopia, eye fatigue and transient refractive changes induced by VDT work in healthy subjects [25]. All these results suggest the possibility of preventing VDT work-induced asthenopia through the intake of some specific nutrients.

The beneficial antioxidant, anti-inflammatory, neuroprotective, immunomodulating and adaptogenic effect of the molecules examined in our study was demonstrated on the retina, vitreous and ocular surface [26]. Previous studies [27,28,29] have shown that dietary modifications and dietary supplements can be used to prevent and/or treat ocular surface conditions. Micronutrients from both food intake and nutraceuticals can affect the morphology and function of ocular surface components through various metabolic pathways, resulting in improved quality of vision and also quality of life.

On the other hand, the main limitation of all these studies on nutritional supplementation is the lack of plasma level dosage before and after their intake. Data can be influenced not only by basic diet and dietary practices (vegetarian, vegan, etc.), but also by the length of the nutraceutical intake period [29].

Regarding the quality of life examined with CVSS: after 1 month of oral intake of Meramirt CM^®^, the score significantly improved and, with it, the visual discomfort of the subjects; a relationship was demonstrated between the quality of life and the amount of use of the screen [11]. The oral intake of the substances studied was able to improve and normalize the perception of the quality of life in our sample, probably due to the widely accepted effects of the components of the product studied by us, which are recognized as adaptogens and stimulants of the metabolism.

Our data are promising but have two main limitations: the short follow-up and the small sample size.

Further observations are needed to confirm these preliminary data and to evaluate the cost-effectiveness of using nutraceuticals in patients who spend many hours using screens, bearing in mind that many of the visual symptoms experienced by VDT users are only temporary and usually decrease after interrupting computer work or using digital devices.

The first thing to do when examining VDT workers who report a reduction in vision is to identify the cause of the problem since a physical problem (such as the presence of dry eye, blepharitis, incorrect refractive error) must be corrected to avoid the onset of symptoms that can become more difficult to eliminate.

Moreover, all visual problems associated with computer vision syndrome have to be investigated and reduced by controlling the lighting and glare on the screen, establishing correct working distances and posture for viewing the screen and ensuring that even minor vision problems are properly corrected to avoid ocular and postural problems.

When eye strain and its consequences are present despite all these precautions, then the present study has shown that some nutraceuticals may improve symptoms due to prolonged use of VDT and related quality of vision. These observations can be generalized to disorders that have arisen in many people who are not VDT workers but who, due to the COVID-19 pandemic, have had to spend a lot of time on VDT, developing symptoms and signs of computer vision syndrome.

The intake of these nutraceuticals, improving the quality of vision and reducing the symptoms associated with the use of VDT could have indirect advantages on the concentration and work productivity of those who take them.

## 5. Conclusions

The intake of some nutraceuticals is able to improve contrast sensitivity in people who spend many hours in front of the PC screen, reducing the symptoms of discomfort and positively influencing their perception of quality of life, as revealed in the CVSS questionnaire.

The fact that both quality of vision and symptoms of computer vision syndrome improved, are two factors that can potentially lead to better work performance and a reduction in disturbances related to computer work and lost working days due to eye disorders.

## Figures and Tables

**Figure 1 nutrients-13-04449-f001:**
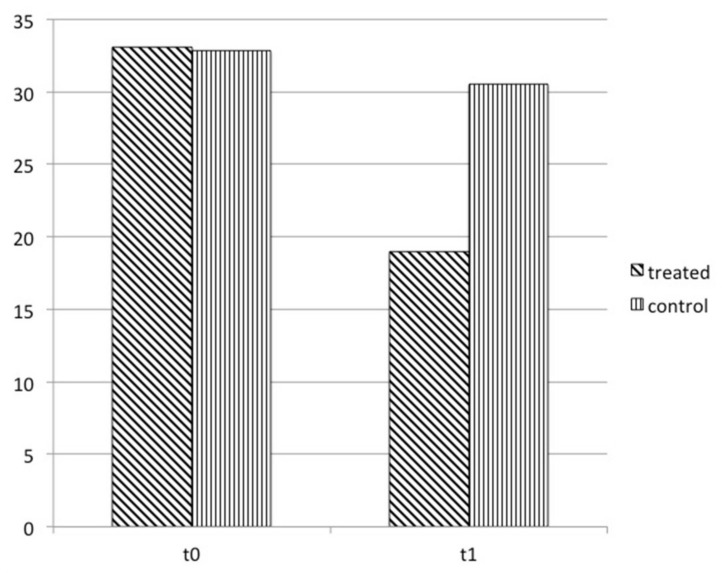
CVSS17 questionnaire score in treated group and in control at enrollment and 1 month later. The score significantly improved from t0 to t1 in treated.

**Table 1 nutrients-13-04449-t001:** Demographics at baseline.

	Treated (T)	Control (C)
Age (years, mean ± SD)	44.3 ± 9.5	45.6 ± 8.7
Gender (women, *n*, %)	11 (77.3)	12 (80)
Schooling level (*n*, %)		
- secondary school	4 (26.7)	6 (40)
- high school	8 (53.3)	6 (40)
- bachelor	3 (20)	3 (20)
Daily time spent at VDT (hour.minute)	7.30 ± 15	7.15 ± 30
Systemic diseases		
- systemic hypertension	3 (20)	2 (13.3)
- thyroid disorders	1 (6.6)	1 (6.6)
- venous disorder	-	1 (6.6)
- anxiety	1 (6.6)	-

SD: Standard deviation.

**Table 2 nutrients-13-04449-t002:** Contrast sensitivity changes over one month period in subjects receiving a tablet a day of Meramirt CM (treated, T) and in a control group (control, C). (+ = improved examination; − = worsened examination).

Spatial Frequency (Cycles per Degree, cpd)	Normal Values (Range)	Contrast Sensitivity t0	Contrast Sensitivity Variation t1	*p*
Treated (T)				
1.5	30–120	52.3 ± 16.6	+37.7 ± 15.1	0.012
3	40–170	78.3 ± 25.2	+27.7 ± 15.1	0.052
6	50–180	46.5 ± 9.9	+6 ± 8.5	n.s.
12	20–130	18.7 ± 2.6	+16.5 ± 2.5	0.010
18	5–70	12.2 ± 1.8	+15 ± 1.7	0.000
Control (C)				
1.5	30–120	65.4 ± 17.6	−2.37 ± 16.5	n.s.
3	40–170	101.2 ± 36.6	+8.1 ± 9.2	n.s.
6	50–180	69.8 ± 9.5	−4.7 ± 4.5	n.s.
12	20–130	29.5 ± 6.5	−1.5 ± 7.2	n.s.
18	5–70	12.5 ± 2.3	+4.1 ± 2.8	n.s.

## Data Availability

Data supporting reported results can be found in a dataset generated during the study.

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
