# Peer review of "A Pilot, Phase II, Observational, Case-Control, 1-Month Study on Asthenopia in Video Terminal Operators without Dry Eye: Contrast Sensitivity and Quality of Life before and after the Oral Consumption of a Fixed Combination of Zinc, L-Carnitine, Extract of Elderberry, Currant and Extract of Eleutherococcus."

_nutrients, 2021, doi:10.3390/nu13124449_

Round 1

Reviewer 1 Report

  1. Thorough proof-reading for spelling grammar and sentence structure is needed. For example:
  • Page 1, line 36 should be changed to “The prolonged use of video terminals not only generate muscolo-skeletal disorders, but also cause ocular complications
  • Line 38, change “pandemy” to pandemic
  • Line 164, correct “Fifhteen” to “Fifteen”
  1. Page 3 line 135to 150 should be moved to an appropriate point in the discussion instead of the materials and methods section.
  2. In the results section, a table for subject demographics would suit better
  3. Was the observer masked? If not, then once again the study results are highly subject to bias.
  4. Discussion line 196 “Is it possible to improve the quality of vision and also the quality of life of video terminal workers with the intake of some nutraceuticals with specific effects on sight? Our data should answer yes” This line sounds like advertorial rather than an authentic research article.
  5. Considering that this was open label study with no placebos, randomization or masking, self-reported symptoms questionnaires are subject to a high degree of bias. Moreover, the sample size is very small and only a minimal statistical significance was observed for visual acuity by an unmasked observer. Therefore, the authors should tread with caution in interpreting the data and refrain from making any sweeping statements.

Reviewer 2 Report

I read the manuscript with interest as the design of experiment was very close to real life. It would be nice to review the manuscript carefully and correct typo. In addition, if there was a positive treatment group with lutein or zeaxanthin. 

Round 2

Reviewer 1 Report

Professional proof-reading and editing for English Language style is recommended

Author Response

Dear Reviewer

the manuscript has been revised for style and English editing